# Fast-Rising Electric Pulses by Reducing Membrane Tension for Efficient Membrane Electroporation

**DOI:** 10.3390/membranes15050151

**Published:** 2025-05-16

**Authors:** Ping Ye, Lulu Huang, Kuiwen Zhao

**Affiliations:** School of Health Science and Engineering, University of Shanghai for Science and Technology, Shanghai 200093, China; iamyeping@usst.edu.cn (P.Y.); 223332599@st.usst.edu.cn (L.H.)

**Keywords:** molecular dynamics, electroporation, rise time, phospholipid membrane, surface tension

## Abstract

Membrane electroporation is an emerging minimally invasive ablation technique being rapidly applied in the ablation treatment of tumors and heart conditions. Different rise times of electric fields lead to variations in the distribution and duration of electric field strength on the cell membrane. This study investigated the effect of the electric field’s rise time on membrane electroporation characteristics using molecular dynamics simulations. The results showed that fast-rising electrical pulses can significantly reduce the membrane tension induced by the Coulomb force within a short period of time and lead to a trend of the electric field angle distribution towards smaller values below 45°, thereby effectively promoting the pore formation process. Optimizing the electric field’s rise time is an effective electroporation ablation strategy, potentially improving the efficacy of clinical cancer treatment.

## 1. Introduction

Electroporation, as a non-thermal ablation modality, has demonstrated significant advantages in tumor treatment and cardiac ablation in recent years. It induces the formation of transient nanopores in the cell membrane by applying a high-intensity electric field, leading to the irreversible electroporation and subsequent apoptosis of target cells while avoiding the side effects of thermal injury to surrounding tissues [1,2,3,4,5,6,7]. Molecular dynamic (MD) simulations have provided a novel perspective for the study of membrane electroporation, serving as an important complement to existing experimental methods. The molecular mechanisms of the cell membrane under the action of electric pulses, including the formation of water pores in lipid bilayers, have been elucidated [8]. Although preclinical studies have confirmed the effectiveness of this technology, optimizing electric field parameters to achieve efficient electroporation and enhance pulse ablation efficiency remains a central challenge in current research.

In practical electroporation ablation experiments, pulsed electric fields are typically employed to interact with the cell membrane. Studies have shown that different pulse parameters (including millisecond, microsecond, and nanosecond pulses) significantly regulate the dynamic process of cell death. Among them, the optimization of pulse duration and pulse type can directly affect the activation degree of immune response, thereby contributing to the improvement of therapeutic effects [9,10]. Different rise times of the electric field can lead to changes in the distribution of the electric field on the cell membrane and the duration of its action, thereby affecting the outcomes of membrane electroporation. However, traditional electroporation research has primarily focused on optimizing basic parameters such as electric field strength and pulse duration, while the correlation mechanisms between the dynamic characteristics of the electric field (such as rise time and waveform changes) and membrane electroporation efficiency remain underexplored. Recently, studies have indicated that the temporal characteristics of the electric field waveform may significantly influence electroporation efficiency by modulating the accumulation process of the transmembrane potential [11]. Other studies have elucidated the complex behavior of pore formation in giant unilamellar vesicles under different loading rates in aspiration experiments and predicted that the behavior of planar phospholipid membranes under the action of an electric field is similar to that under lateral tension [12,13]. Specifically, relevant studies have demonstrated that the prolongation of a pulse’s raised time can lead to an increase in the critical voltage value required for the formation of the first membrane pore, thereby affecting the efficiency of the electroporation process [14]. It is worth noting that the rapid changes in the electric field during the rise and fall of the pulse amplitude were the cause of intracellular effects [15]. Single pulses with short rise and fall times can induce electric field-dependent increases in calcium ion influx, the dissipation of mitochondrial membrane potential, and cell death [16]. These findings highlight the importance of optimizing the rise and fall times of the electric field for improving electroporation efficiency and controlling intracellular effects.

This study analyzed the effects of different electric field rise times on the cell membrane using molecular dynamics simulations. The results showed that reducing the electric field’s rise time significantly decreased the membrane tension caused by the Coulomb force, which, in turn, facilitated the reorientation of phospholipid molecules and, thus, promoted pore formation. Specifically, rapidly rising electrical pulses can cause the electric field angle distribution to more readily favor values below 45° within a shorter period of time, thereby more effectively promoting pore formation. In clinical tumor ablation therapy, optimizing the electric field’s rise time can serve as an effective electroporation strategy.

## 2. Materials and Methods

The phospholipid membrane, composed of 1024 1,2-dipalmitoylphosphatidylcholine (DPPC) molecules, was constructed by the CHARMM-GUI membrane constructor [17,18]. DPPC, a representative phospholipid in mammalian cell membranes, has been widely used in studies of membrane pore dynamics [19,20,21]. The phospholipid membrane was immersed in a solution composed of 244 sodium ions, 244 chloride ions, and approximately 21,000 water molecules, with a 0.15 M NaCl concentration, which closely mimics the ion composition of extracellular fluid in physiological conditions. Figure 1 depicts the simulated system at a box size of approximately 17.3 nm × 17.3 nm × 12.5 nm. All systems were simulated using the GROMACS 2023.5 simulation package [22] at coarse-grained (CG) resolution using the Martini 2.2 force field [23,24,25]. The Martini force field employs a “four-to-one” coarse-graining strategy, mapping an average of four heavy atoms and their bonded hydrogen atoms to a single coarse-grained bead. Each DPPC molecule is composed of 12 Martini particles: the hydrophobic tail chains of sn-1 and sn-2 are each represented by four hydrophobic beads (C1B–C4B and C1A–C4A, respectively); the polar head includes a negatively charged phosphate group (PO4) and a positively charged choline group (NC3); the glycerol backbone is constructed by two neutral beads (GL1 and GL2), with GL1 bridging the phosphate group and the sn-2 chain, and GL2 linking GL1 to the sn-1 chain. A periodic boundary was applied in three directions [20]. The leap-frog algorithm was utilized in this MD simulation to integrate the equations of motion. The temperature was maintained at 310 K using the Berendsen algorithm [26], and a pressure of 1 bar was coupled semi-isotropically using the Parrinello–Rahman algorithm [27]. Short-range electrostatic and Lennard–Jones interactions were cut off at 1.2 nm, and long-range electrostatics were calculated by the PME algorithm [28]. All simulations utilized a 20 fs MD timestep [29], with the coordinates saved every 0.1 ns. Once the equilibrium configuration was reached (after 100 ns), MD simulations were performed, with production run times of 10 ns at 0.16 V/nm and 50 ns at 0.18 V/nm and 0.20 V/nm.

In this study, the electric field applied to the simulated system was divided into two stages: the rising phase and the constant phase (Appendix A). During the rising phase, the electric field that increased linearly with time along the Z-axis was applied, with the field strength gradually increasing from zero to the preset peak value. In our study, the rise times of the electric field were set at 1 ns, 5 ns, 10 ns, 15 ns, 20 ns, and 30 ns. Once the electric field reached the predetermined peak value, it entered the constant phase, during which a constant electric field equal to the peak value was applied until pore formation was observed. Under the same peak electric field conditions, the maximum electric field strength corresponding to different rise times was consistent and equal to the peak electric field strength. For each electric field configuration, the simulation was repeated five times.

The peak electric field strengths selected for this study were 0.16 V/nm, 0.18 V/nm, and 0.2 V/nm, which are an order of magnitude higher than the typical electric field strength of nanosecond pulsed electric fields in experimental studies [30,31]. In current molecular simulations of membrane electroporation, the applied electric field strength is generally higher than experimental standards, regardless of whether coarse-grained methods or all-atom force fields are used [11,32,33,34]. Additionally, under the condition of a peak electric field of 0.16 V/nm, following the constant electric field phase, we conducted simulations with falling edge durations of 1 ns, 5 ns, and 10 ns to investigate whether pore recovery occurred.

To quantitatively analyze the orientation characteristics of the phospholipid membrane under different electric field rise times, this study employed an angle analysis method based on spatial distribution. The simulation box was divided into 1 nm × 1 nm subregions along the X and Y axes. The angle between the phospholipid membrane’s normal and the Z-axis was calculated within each subregion and the proportion of subregions with angles less than 45° was statistically analyzed. This proportion reflects the spatial orientation characteristics of the phospholipid membrane, which was used to assess its tilting degree relative to the direction of the electric field. All molecular images were generated using the visualization software VMD 1.9.4a57 [35]. Pore formation was identified by the formation of a continuous water chain spanning the hydrophobic core of the phospholipid membrane [36]. The analysis of the simulation trajectories was completed using the built-in analysis tools in the GROMACS 2023.5 software package. The membrane surface tension was computed from the pressure tensor obtained via the “gmx energy” command, following Equation (1) [37].(1)$$\gamma =\frac{1}{2}L_{z}\left[P_{zz}-\frac{1}{2}(P_{xx}-P_{yy})\right]$$

Here, $L_{z}$ is the dimension of the simulation box in the Z-axis, with $L_{z}$ = 12.5 nm in this study; $P_{zz}$ is the pressure component perpendicular to the membrane surface (in the Z-axis direction); and $P_{xx}$ and $P_{yy}$ are the pressure components parallel to the membrane surface (in the X and Y axes).

## 3. Results and Discussion

### 3.1. Fast-Rising Electric Pulses Promote Membrane Electroporation Effects

In practical electroporation applications, electric pulses are typically used instead of constant electric fields. To make the simulation experiments more realistic, this study systematically investigated the effects of electric pulses with different rising-edge characteristics on phospholipid membrane electroporation by adjusting the electric field’s rise time.

We applied a linearly rising electric field along the Z-axis, increasing from zero to the preset peak value. Once the peak value was reached, the electric field strength was maintained until pore formation occurred. In this study, the pore sizes formed during the application of the constant electric field had already approached the boundary of the simulation box. Consequently, under the peak electric field of 0.16 V/nm, no pore recovery was observed in the simulations with falling edges of 1 ns, 5 ns, and 10 ns conducted after the constant electric field phase. However, pore recovery might be observed is the falling edge is increased for smaller pores. As shown in Figure 2, the pore formation time was significantly affected by the electric field’s rise time, with a shorter rise time leading to a shorter pore formation time. Pore time refers to the total time from the beginning of the simulation to the formation of the pore, including the duration of the electric field’s rise phase. As the electric field’s rise time decreased from 30 ns to 1 ns, the pore formation times for electric pulses with peak field strengths of 0.16 V/nm, 0.18 V/nm, and 0.20 V/nm were shortened by 2.01, 3.28, and 5.81 times, respectively. This result is consistent with previous studies [38], which have shown that a rapid increase in transmembrane voltage can more easily induce pore formation. It should be noted that, with a 30 ns rise time, the pore formation time under a 0.2 V/nm electric field (31.2 ns) was 0.4 ns longer than that under a 0.18 V/nm electric field (30.8 ns), which may be attributed to the resonance effect between water molecules and the electric field [11]. Additionally, due to the small electric field strength interval in this study, the influence of adjacent high electric field strengths on the pore formation time becomes less significant [39]. The rapid change in the electric field accelerated the polarization process of the membrane, leading to an exponential increase in membrane capacitance, which, in turn, destabilized the membrane structure and facilitated pore formation. When the peak electric field increased from 0.16 V/mm to 0.20 V/mm, the pore formation time also showed a significant downward trend. As described in a previous study, the electric field strength is nonlinearly and positively correlated with the pore formation rate [39]. Additionally, we analyzed the pore formation time starting from the peak electric field (Appendix A). Compared with simulations of longer electric field rise times, simulations of shorter electric field rise times presented longer pore formation times calculated after the electric field peak. Additionally, under lower peak electric field conditions (e.g., 0.16 V/nm), the difference in pore formation times after the peak between fast and slow electric field rise times was smaller than that under higher peak electric field conditions. Considering the membrane deformation during the rapid change in the electric field, the rise time of the electric field should be taken into account when calculating the pore formation time. Specifically, compared with the application of a constant low-intensity electric field [40], electric pulses with a rapid rising edge can more effectively induce pore formation within a shorter period of time. This phenomenon indicates that the rising edge of the electric pulse plays a significant role in enhancing the electroporation effect, thus verifying the potential of fast-rising electric pulses in improving electroporation efficiency.

As shown in Figure 3, we used the VMD top-down view to display the pore structures formed under electric pulses with varying rise times at a peak electric field of 0.16 V/nm. The results show that only one pore forms per cycle in the simulation box under this electric field. As the rise time decreases from 10 ns to 1 ns, the pore formation rate accelerates, and the pore size increases. Notably, when comparing 15 ns, 20 ns, and 30 ns pulses, shorter rise times significantly reduce the time required to form a specific pore size.

### 3.2. The Reverse Effect of Anisotropic Coulomb Forces Induced by Changing Electric Fields on Membranes

The anisotropic Coulomb force induced by the electric field plays a significant role in the electroporation process. When the electric field is applied in the Z-axis, analyzing the forces on the membrane in the XY direction helps to reveal how the electric field’s rise time can affect local membrane deformation and pore formation. To more accurately predict the effects of the electric field’s rise time on membrane electroporation characteristics, we analyzed the Coulomb force on the phospholipid membrane along the XYZ dimensions of the simulation box under different electric field rise times. The Coulomb force in the XY plane was obtained by calculating the vector sum of the forces in the X and Y axes. As shown in Figure 4, the changing electric field rise times exhibited significant anisotropic regulation of the spatial distribution of the Coulomb force experienced by the phospholipid membrane. The electric field’s rise time increased, and the Coulomb force experienced by the membrane in the XY plane showed a clear upward trend (Figure 4a), while the force in the Z dimension exhibited a counteracting weakening effect (Figure 4b). This inverse effect of the anisotropic Coulomb force can alleviate the perturbation of the membrane, thereby further stabilizing the membrane structure and ultimately inhibiting the electroporation process [40]. When the peak electric field was 0.16 V/nm, the electric field force experienced by the phospholipid membrane in the XY plane, which was perpendicular to the membrane normal vector, increased by 8.35 × 10^3^ kJ/mol/nm as the electric field’s rise time increased from 1 ns to 30 ns. Meanwhile, the Coulomb force in the Z-axis, which is parallel to the membrane vector’s normal, decreased by approximately 62%. Notably, as the phospholipid membrane lay in the XY plane, the Coulomb interactions in the XY plane were influenced by both the electric field and internal membrane interactions, resulting in a cumulative effect. By contrast, the Coulomb interactions within the membrane in the Z direction were negligible. Thus, the Coulomb forces in the XY plane were much larger than those in the Z direction. Although the force variation in the XY direction was only about 3%, the change in pore formation time was approximately threefold. This result indicates that even small changes can significantly affect the behavior of the phospholipid membrane, similar to how minor variations in membrane properties can lead to substantial changes in lipid conformation and orientation [41]. Similar results were also observed at peak electric fields of 0.18 V/nm and 0.20 V/nm (Appendix A). This phenomenon indicates that a rapid change in the electric field can significantly weaken the interactions between phospholipid molecules within a short period of time, making the membrane structure more susceptible to perturbation and, thus, facilitating pore formation. These findings provide a new perspective for understanding the mechanism of interaction between the electric field and the membrane and offer a theoretical basis for optimizing electroporation techniques.

### 3.3. Dynamic Electric Field Regulation of Membrane Tension Related to the Electric Field Angle

Under the influence of the dynamic electric field, the structure of the phospholipid membrane undergoes a certain degree of dynamic fluctuation. To further investigate the mechanism by which the electric field’s rise time affects the anisotropic characteristics of the forces on the phospholipid membrane and to verify whether this anisotropy originates from the inhomogeneity of the electric field direction acting on the phospholipid membrane, in particular, we employed the method based on our previous research on the electric field direction [40]. Specifically, we calculated the angle between the phospholipid membrane’s normal and the Z-axis of the simulation box to analyze the orientation distribution characteristics of the angles in different regions of the phospholipid membrane. It should be noted that the phospholipid membrane’s normal is perpendicular to the phospholipid membrane plane, and in this study, the phospholipid membrane’s normal direction corresponds to the Z-axis of the simulation box. In this process, we quantified the tilt angle of the phospholipid membrane relative to the electric field direction by calculating the proportion of regions with angles less than 45°, thereby evaluating its promoting effect on pore formation. This approach allowed for a clearer revelation of the mechanisms by which different electric field rise times affect the electroporation effect on the membrane. To calculate the proportion of regions with angles less than 45°, the phospholipid membrane was evenly divided into 1 nm × 1 nm unit areas along the X and Y axes. With the center of each unit area as the reference, the membrane normal perpendicular to the area was determined, and the electric field angle was defined as the angle between the normal and the Z-axis. The proportion of regions with angles less than 45° was analyzed by tallying the electric field angles of all unit areas. Figure 5 illustrates the tilting of the phospholipid membrane and the Coulomb force in the XY plane under different electric field angles. As the electric field angle increased, the membrane tilting intensified, and the Coulomb force in the XY plane also increased. With a larger angle, the lateral force perpendicular to the membrane’s normal was strengthened, which hindered pore formation.

The variation in the proportion of regions where the angle between the normal of the phospholipid membrane and the Z-axis of the simulation box is less than 45°, where the electric field’s rise time under different peak electric field strengths is shown in Figure 6a. Under the influence of electric fields with different rise times, the phospholipid membrane fluctuated in response, causing the membrane’s normal in different regions to align at varying angles with the electric field direction. When the electric field’s rise time increased from 1 ns to 30 ns, the proportion of regions where the angle between the normal of the phospholipid membrane and the Z-axis of the simulation box was less than 45° and decreased by 12.8% under the peak electric field strength of 0.16 V/nm. Under peak electric field strengths of 0.18 V/nm and 0.20 V/nm, the proportion decreased by 20.5% and 16.5%, respectively. These results indicate that compared with long-rise-time electric fields, short-rise-time electric fields are more inclined to form a smaller angle distribution of the electric field, thereby creating more favorable conditions for pore formation. This is closely related to the reduction in membrane surface tension under low-angle electric fields [40]. As the electric field angle decreases, the membrane surface tension is reduced, making pore formation more readily achievable.

To investigate how the electric field’s rise time affects the characteristics of membrane electroporation, particularly whether it modulates membrane surface tension by altering the angle distribution of the electric field acting on the phospholipid membrane, we analyzed changes in membrane surface tension under electric pulses with different rise times. The electric field angle analysis used the instantaneous configuration at the end of the electric field’s rise phase as the reference point, directly measuring the distribution of the phospholipid membrane’s normal in relation to the simulation box. The results showed that as the electric field’s time to rise decreased, the membrane tension exhibited a significant downward trend (Figure 6b). This further confirmed that the adjustment of the electric field’s rise time was effective in changing the electric field angle distribution by affecting membrane tension, thereby regulating the characteristics of membrane electroporation. Under the influence of a rapid electric field rise, the phospholipid headgroups, with their short rotational time constant (0.1–0.5 ns) [42], can quickly respond to the electric field changes and adjust their orientation to facilitate pore formation. Even with a low external electric field strength, nanosecond electric field pulses can significantly enhance membrane perturbation efficiency. Therefore, we recommend optimizing electroporation characteristics in experimental design by adjusting the rise time of the electric field pulse rather than simply increasing the field strength. However, due to the relatively small differences among the three selected peak electric field strengths (0.16 V/nm, 0.18 V/nm, and 0.20 V/nm), the changes in phospholipid membrane orientation and membrane tension did not yet show a clear trend with variations in the peak electric field strength. In short, under the action of a fast-rising electric pulse, the degree of membrane deformation was relatively small, and the proportion of regions where the angle between the membrane’s normal and the electric field direction was less than 45° increased. This distribution of electric field angles made the electric field direction nearly perpendicular to the membrane surface during the constant phase, thereby increasing the Coulomb force of the membrane in the Z-axis and decreasing it in the XY plane. This indicates that the smaller membrane deformation induced by rapidly rising electrical pulses resulted in lower membrane surface tension, thereby accelerating pore formation and enlarging pore size. Conversely, the significant membrane deformation caused by slow-rising electrical pulses increased membrane surface tension, requiring more energy to overcome the barrier for pore formation and, thus, necessitating the longer duration of electric field exposure. This finding revealed the significant role of the electric field’s rise time in regulating membrane tension and pore formation, providing an important theoretical basis for optimizing electroporation techniques.

## 4. Conclusions

This study, through molecular dynamics simulations, reveals the significant impact that the electric field’s rise time has on membrane electroporation characteristics, offering new insights into the molecular mechanisms of electroporation. The findings indicate that adjusting the electric field’s rise time can significantly alter membrane tension, thereby regulating the orientation of phospholipid molecules and the kinetics of pore formation. Specifically, the rapidly rising electric pulse drove the electric field direction to be more perpendicular to the membrane surface, significantly reducing electric field angle-dependent membrane surface tension over an extremely short time, thereby effectively promoting pore formation. In clinical tumor and cardiac ablation therapy, optimizing the rise in the electric field’s time can serve as an effective electroporation strategy and lay a theoretical foundation for optimizing electroporation techniques and developing personalized treatment plans in clinical applications.

## Figures and Tables

**Figure 1 membranes-15-00151-f001:**
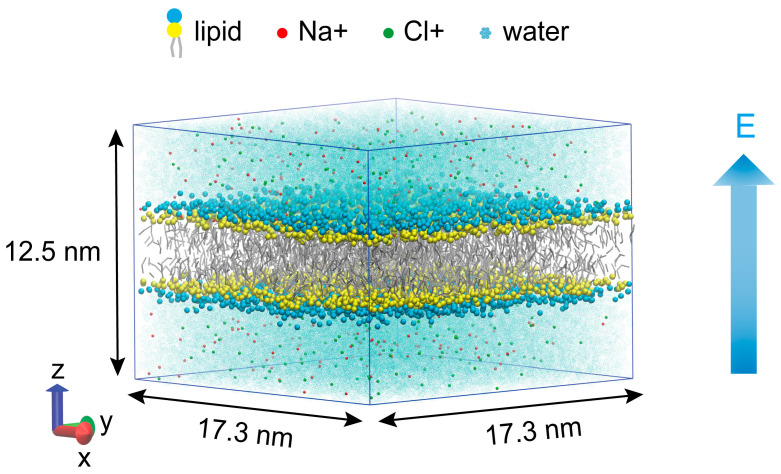
Side view of the phospholipid membrane system. Dark blue, yellow, and grey particles represent the DPPC phospholipid membrane, and the light blue, red, and green particles represent polar water molecules, sodium ions, and chloride ions, respectively. The external electric field applied along the Z-axis.

**Figure 2 membranes-15-00151-f002:**
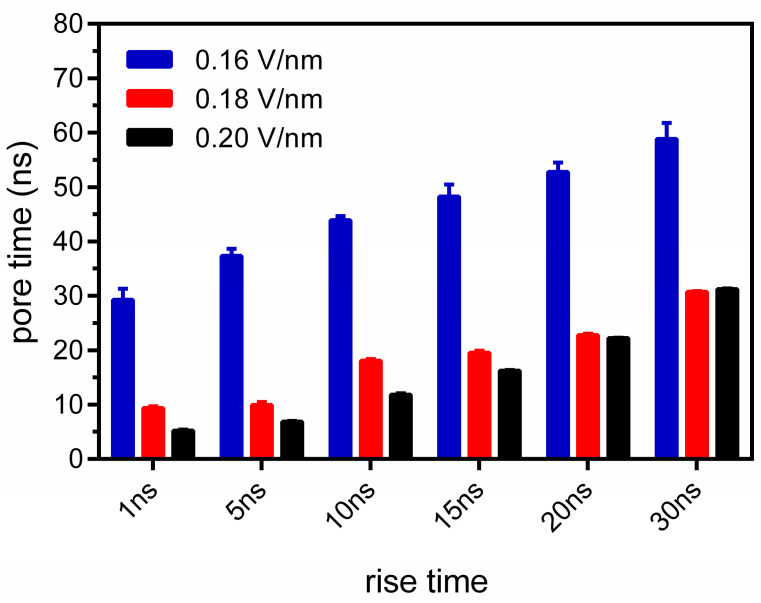
The influence of the electric field’s rise time on the pore formation time. The red, blue, and black bars represent the peak electric field of 0.16 V/nm, 0.18 V/nm, and 0.20 V/nm, respectively. The error bars represent the standard deviation of the five repeated simulations.

**Figure 3 membranes-15-00151-f003:**
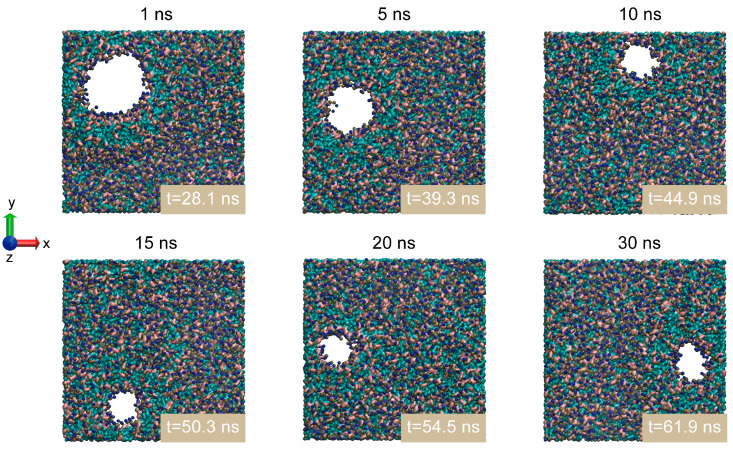
Snapshots of pore formation in the phospholipid membrane with varying electric field rise times (peak electric field: 0.16 V/nm). Here, t represents the pore formation time, including the electric field’s rise time.

**Figure 4 membranes-15-00151-f004:**
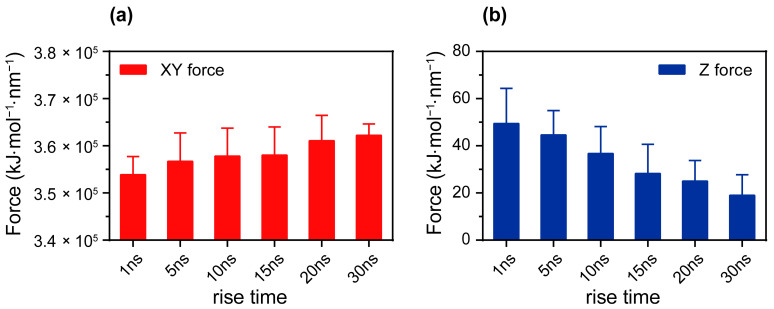
The Coulomb force acted on the phospholipid membrane via an electric field with various electric field rise times (peak electric field: 0.16 V/nm). (**a**) The Coulomb force acting on the phospholipid membrane in the XY plane under different electric field rise times. This Coulomb force was calculated by the vector summation of the X and Y axes. (**b**) The Coulomb force acting on the phospholipid membrane in the Z direction under different electric field rise times. The error bars represent the standard deviation of the data within a 1 ns time window (excluding the electric field rise time phase) centered around the pore formation moment.

**Figure 5 membranes-15-00151-f005:**
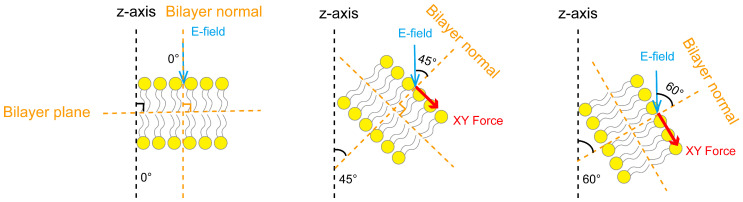
Illustrations of the phospholipid membrane tilting and Coulomb forces in the XY plane under different electric field angles. The electric field angles represent the angles between the electric field direction and the membrane’s normal at 0°, 45°, and 60°. Dashed lines are used as references for the electric field angles. Yellow dashed lines represent the phospholipid membrane plane and the phospholipid membrane’s normal, which are perpendicular to each other; the black dashed line indicates the Z-axis of the simulation box; the blue arrow shows the applied electric field direction, parallel to the Z-axis of the simulation box; and the red arrow indicates the Coulomb force on the membrane in the XY plane, perpendicular to the membrane normal, with the arrow length representing the force magnitude.

**Figure 6 membranes-15-00151-f006:**
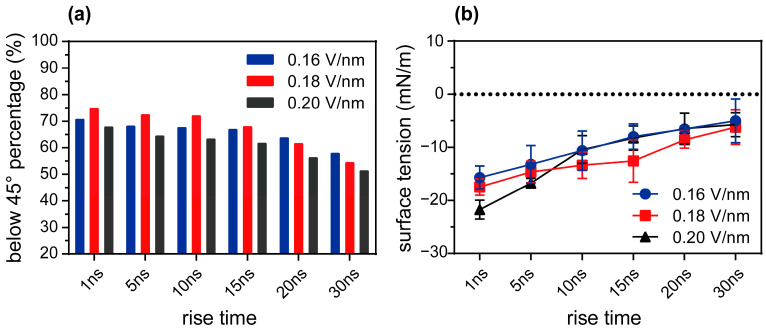
The effects of the electric field’s rise time on phospholipid membrane structure and surface tension. (**a**) Analysis of phospholipid membrane orientation. The simulation box was divided into 1 nm × 1 nm subregions along the X and Y axes. The angle between the membrane’s normal and the Z-axis was calculated in each subregion and the percentage of subregions with angles < 45° was determined. The blue, black, purple, and black bars represent the peak electric field of 0.16 V/nm, 0.18 V/nm, and 0.20 V/nm, respectively. (**b**) The surface tension of the phospholipid membrane induced by the electric field at various rise times. The blue, black, purple, and black lines represent the peak electric field of 0.16 V/nm, 0.18 V/nm, and 0.20 V/nm, respectively.

## Data Availability

The original contributions presented in this study are included in the article/Appendix A. Further inquiries can be directed to the corresponding author.

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
