# Peer review of "Fast-Rising Electric Pulses by Reducing Membrane Tension for Efficient Membrane Electroporation"

_membranes, 2025, doi:10.3390/membranes15050151_

Round 1

Reviewer 1 Report

Comments and Suggestions for Authors

In this work, the authors have studied the electroporation of hydrated bilayers when subjected to electric fields of varying strength. They have specifically focused on the impact of the rise-time parameter and its impact on membrane structure, pore formation, and Coulomb forces. Although this parameter seems to be important for the applicability of this technique in possible therapeutic applications, I feel this study would benefit from additional analysis. Several methodological details are missing, as well as a more in-depth explanation of some of the analyses. To potentially improve the relevance and scope of the work, in my opinion, it would be important to extend this work to other relevant parameters, such as fall-time and electric field width.

Specific comments:

1. In the following sentence, “[…] in the ablation treatment of tumors and the heart.”, the authors should refer the heart condition and not just the organ (page 1, line 9).

2. I do not understand the following statement – “[…] approximately 21.000 water molecules, which is closest to the extracellular fluid of the body.”. Are the authors referring to the number of water molecules or to the concentration of ions in the extracellular fluid (page 2, line 62)?

3. A justification for choosing the DPPC phospholipid should be provided.

4. Authors should provide both the GROMACS version and the actual MARTINI force field that was employed in the study.

5. Why did the authors use the Berendsen thermostat? There are plenty of studies in literature stating its difficulties in producing correct thermodynamic ensembles.

6. In the following statement, “All bonds involving hydrogen atoms were constrained using the LINCS algorithm [24].”, if the authors used CG simulations, where there any hydrogen atoms to constraint (page 2, line 71)?

7. In the following sentence, “[…] extensive MD simulations have been performed.”, the authors should state the duration of the production runs (page 2, line 74).

8. In Table S1, the MD timestep does not seem in line with the 20 fs that were referred to in the methods section. Please revise.

9. Can the authors refer to the experimental standard of electric field strength? It would be advantageous to know how much the experimental and in silico values differ on average. Also, one of the advantages of CG simulations is the accessibility to extended time scales. So, why not explore the effects at lower electric field strengths, closer to the experimental values?

10. I believe a depiction of the cited angle between the normal of the bilayer and the z axis would be helpful, highlighting the different situations, e.g., angle below and higher than 45º and its implications to membrane tilt and organization. In addition, was the onset of pore formation evaluated just by visual inspection? Couldn’t the authors use a more quantitative approach?

11. Figure 2 (page 4) depicts error bars. Did the authors perform replicate simulations? If so, this is not stated in the manuscript. Conversely, I think this is an important practice in any MD simulation study, and this would enhance confidence in the results obtained.

12. On the following statement: “Given that the focus of the study was on the pore formation stage, the falling edge of the electric field was not simulated.” – could the authors provide a prediction, based on available literature, of what they expect would happen at the falling edge of the electric field? Would the membrane recover, and could the field rise impact the recover rate? I think this study would probably benefit from this analysis and would make these simulations more realistic. (page 3, line 112)

13. About the pore formation time, is this counted from the beginning of the simulation, or after the peak electric field. If it is counted since the beginning of the simulation, if there were a more rapid increase of the electric field, I would say it is expected that the time for pore formation would be lower. Time-to-pore after reaching peak field and from field onset should be analyzed. It is not immediately clear which one the authors used.

14. The 0.2 V/nm should be written as 0.20 V/nm for consistency with significant numbers. Also, in the literature, we often find different ranges of electric field values. Some studies have used larger values (> 1.5 V/nm). Could this be related to the different nature of the force fields that were employed? I think it is important that the authors discuss the strength of the electric fields relative to other studies in literature and provide reasoning for their choice.

15. I also feel the employed electric fields were a bit too strong. I think it would be best to use lower field strengths (for example 0.1 V/nm), since pore formation is very fast for the 0.17-0.18 V/nm simulations. Also, the results for 0.17 V/nm and 0.18 V/nm were very similar. Since the authors used CG simulations, I believe extended time scales should be accessible for lower electric field strengths.

16. Further analyses are required, particularly on how the authors have measured pore formation. Some pictures showing the formation and size of the pores would also aid the interpretation of the work. Also, isn’t the size and number of pores a good parameter to assess the dynamics of pore formation using rapid and slow electric field pulses?

17. Regarding Coulomb and Angle analyses, it is not clear how were these measured – after peak onset or before? Also, were they averaged over the simulation time window? If so, there should be a representation of the standard error or standard deviation for these results.

Author Response

Please See the attachment to access the responses. 

Reviewer 2 Report

Comments and Suggestions for Authors

In this manuscript, the authors reported effects of electric pulses on the membrane electroporation, particularly focused on the rising time of the field, using molecular dynamics simulations. Although it is a continuation of the authors' previous work (ref. [32]) and has some overlap of the contents, its focus on the rising phase is new, and hence it is acceptable as a separate paper.

However, there are some inconsistencies and errors in the current text. I would like to ask the authors to correct and/or supplement these points (although some of these are very minor).

Line 60:
"CHARMMGUI" should read "CHARMM-GUI".

Line 65:
Which version of MARTINI force field was used?

Lines 71-72:
If the MARTINI coarse-grained force field was used, the statement "All bonds involving hydrogen atoms were constrained using the LINCS algorithm [24]" is strange; please check.

Lines 66-:
The reference numbers [20]-[25] seem shifted by 1.

Line 72:
The MD time step is 20 fs here, while 2 ps in Table S1 (I think the former is correct, however please confirm).

Lines 86-87:
Here, "a constant electric field equal to the peak value was applied until pore formation was observed", however in Table S1, "Constant time" is 50 or 10 ns, which is sometimes shorter than the "pore time" in the result.
Also, please clarify whether the "pore time" includes the rising phase or only the constant phase.

Lines 95-96:
The definition of "the normal of the phospholipid membrane" is not clear enough.
How to calculate it?

Lines 100-102:\
The definition "the appearance of continuous water molecule diffusion" is also a little bit unclear, though acceptable.

Line 118:
"previous study 9" It seems it is not ref. [9]; please check.

Figure 2:
When the rise time is 30 ns, the pore time of the 0.2 V/nm case seems longer than the 0.18 V/nm case.
Is it just within the fluctuation? Or are there any possible reasons?

Line 132:
There is no "purple" bar.

Lines 150-151:
"the electroporation process 37." It is apparently not ref. [37]; please check.

Figure 3:
In panel (b), the force values (tens of KJ / mol nm) correspond to tens of pN, which is reasonable.
However, in panel (a), the values are 5 orders of magnitude higher (hundreds of nN). Why so large?
Also, the change of the XY force (a) is only 3%. Does it affect the behavior?

Please also clarify at which point the force was measured (averaged over a certain period?)

Lines 167-168:
The statement "The Coulomb force of phospholipid membrane in the Z dimension is parallel to the membrane normal vector under different electric field rise time." is confusing (I think the measured forces were decomposed into (a) and (b)).
(This also applies to Figs. S1 and S2)

Line 170:
"The text continues here." Please delete it.

Line 180:
"we only calculated the proportion of regions with angles less than 45°"
Here also, please clearly state at which point the angles were measured.
Although the percentage of <45-degree regions may be a good index, the entire distribution may help understanding.

Lines 202-204:
"This further confirmed that the adjustment of the electric field rise time is effective in changing the electric field angle distribution by affecting membrane tension, thereby regulating the characteristics of membrane electroporation."
The result itself is fine.
However, if so, the time constant is probably related to the rotational properties of the membrane molecules.
Then, even with a weaker electric field, the characteristic time scale may be still in the nanosecond range.
How can we connect this result to real experimental design? Do you have any ideas?

Line 238 and caption of Figure S1:
"peak electric field: 0.16 V/nm"
It is same as Figure 3. Isn't it 0.18 V/nm?

Line 246:
"Please add:"
Line 253-255:
"In this section, you can acknowledge any support given which is not covered by the author contribution or funding sections. This may include administrative and technical support, or donations in kind (e.g., materials used for experiments)."
Please delete these guide text.

Table S1:
"The electric field scheme of the peak electric field and Z axis at different rise time"
It is difficult to follow. What is Z axis?
MD time step and constant time are inconsistent with the main text.

Comments on the Quality of English Language

There are some errors in the text (not limited to the English), as listed above.

Author Response

Please see the attachment to see the response. 

Reviewer 3 Report

Comments and Suggestions for Authors

the paper is interesting and ddescribes a topic with high impat in the comprehension of electroporation. It has to be improved.

some suggestions to improve the paper: in introduction please add references related to main paper in simulationof membrane electroporation.e.g. check authors as Kotnik or Miklavcic.
in material and methods please, describe accurately the output of the referred models and software. please add also equations.
In experimetal design please, clarify if the maximum electric field strength is the same for all the rise time. specify this value.
please, al linee 270-272 revise  the  sentence. It is not clear.
Please, specify all acronyms.

Author Response

We would like to express our deepest appreciation for the positive feedback and insightful comments provided by the reviewers, which have been of immense help in refining our manuscript. We have revised the manuscript following the reviewers comment (main revisions in the revised manuscript are highlighted in red font) and made the following point-by-point replies to the specific comments. The detailed response is provided in the accompanying attachment "Response to Reviewer3".

Round 2

Reviewer 1 Report

Comments and Suggestions for Authors

1. In the following sentence: "Once the equilibrium configuration was reached (after 100 ns), extensive MD simulations have been performed with production runs lasting from 10 to 50 ns.", authors should indicate which ones were run for 10 (0.18 and 0.20 V/nm) or 50 ns (0.16 V/nm). I would also avoid the term "extensive".

2. From what I gathered, higher electric field strengths are typically employed to accelerate electroporation events in MD simulations (either CG or AA). The authors should indicate that this is common practice, when they discuss "real" and simulated electric field strengths.

3. I do not understand the author's reasoning in this sentence: "Our study focused on irreversible electroporation, where the plateau phase of the electric field is sufficient to cause irreversible damage to the cell membrane, and therefore, no pore recovery was observed even with an extended fall time.". I understand that they want to focus on irreversible electroporation, but I do not understand their claim in the fall time. Did the authors test or not the fall time parameter? If so, this is not stated in their protocol. Also, I would assume that pore recovery would be observed, specifically for the cases where a smaller pore was formed.

4. Regarding the pore formation time after peak onset, I would not say there was no significant trend. It is not clear if the authors evaluated this for all replicas (if so, they should provide some error values), but it seems that for faster rise time simulations, pore formation after peak onset is slower than for slower rise times. And I would assume that for lower electric field strengths (e.g., 0.16 V/nm) the difference between fast and slow rise times would be smaller.

5. Maybe there is some confusion in my part, but if regions below 45º are indicative of more ordered membranes, shouldn’t the 1 ns rise time have a lower percentage of these regions, because pores are bigger? In summary, I would assume that if we have a more negative surface tension in the membrane at faster rise times, the lateral pressure would be higher, and we would have seen more membrane deformation that could enhance pore formation. But I would assume that the regions with angles below 45º would be lower compared to the higher surface tension situations. Can the authors clarify? Maybe I am mixing the tilt induced by pore formation (near the pore), with the ordering of the whole membrane due to compression. I would also assume higher XY Coulomb forces for membranes with a negative surface tension.

Other than these points, the authors have responded to most of my questions, and I would suggest publication after minor revision.

Author Response

We would like to express our deepest appreciation for the positive feedback and insightful comments provided by the reviewers, which have been of immense help in refining our manuscript. We have revised the manuscript following the reviewers comment (main revisions in the revised manuscript are highlighted in red font) and made the following point-by-point replies to the specific comments. The detailed response is provided in the accompanying attachment "Response to Reviewer1".

Reviewer 3 Report

Comments and Suggestions for Authors

no more  comments